# Negative health effects of dental X-rays: A systematic review

**Laila Wiklander**[1,2]*, **Andreas Cederlund**[1,2], **Nils Kadesjö**[2,3], **Peggy Näsman**[4], **Sofia Tranaeus**[5], **Aron Naimi-Akbar**[2,5]

**1** Department of Maxillofacial Radiology, Folktandvården Eastmaninstitutet, Public Dental Health, Stockholm, Sweden, **2** Department of Dental Medicine, Karolinska Institutet, Stockholm, Sweden, **3** Department of Medical Radiation Physics and Nuclear Medicine, Karolinska University Hospital, Stockholm, Sweden, **4** Department of Dental Medicine, Division of Cariology and Endodontics, Karolinska Institutet, Stockholm, Sweden, **5** Malmö University, Faculty of Odontology, Health Technology Assessment – Odontology (HTA-O), Malmö, Sweden

* laila.wiklander@ki.se

## Abstract

### Background

This study evaluates whether exposure to dental X-ray examinations in childhood and adolescence results in negative health effects.

### Material and methods

This systematic review includes both primary studies and systematic reviews available in Medline, Embase, and Web of Science databases. Six reviewers read the full text of the selected studies.

### Results

The literature search resulted in 10,949 publications. After title and abstract screening, 55 publications were selected for full text reading, resulting in a total of 18 reports, 7 systematic reviews, and 11 primary studies for quality assessment.

### Conclusion

None of the selected studies passed the quality assessment due to high or very high risk of bias. There is a gap of knowledge regarding negative effects of dental X-rays and a need for more accurate and updated studies.

## Introduction

Dental X-rays are a common procedure in dental care as a part of a clinical examination. Intraoral and extraoral examinations can be used to diagnose cavities, periodontal disease, and pathological problems or to evaluate disturbance in tooth

**Data availability statement:** All relevant data are within the paper and its Supporting Information files

**Funding:** Laila Wiklander receives salary from Folktandvården/Public Dental Health Care, Stockholm County, LLC, Region Stockholm, both as a doctoral student and senior consultant. Folktandvården funded the language revision of the original manuscript. The specific roles of these authors are articulated in the 'author contributions' section. The funder had no role in study design, data collection analysis, decision to publish or preparation of the manuscript.

**Competing interests:** The authors have declared that no competing interests exist

development and growth. Between 2008 and 2022, annual dental radiology examinations increased from approximately 480 million to 1.1 billion [1]. This increase highlights the importance of studying the effects of radiation in dentistry. Intraoral radiographs – i.e., bitewing projections – are the most common radiographs for caries detection [2]. Panoramic and lateral cephalometric radiographs are extraoral x-ray examinations, which are often needed for orthodontic evaluation. Cone Beam Computed Tomography (CBCT) is another type of extraoral radiographic examination in dentistry which over the last 20 years has become more common, as an alternative and complement to the traditional examinations.

Optimizing and justifying every X-ray examination as well as keeping up with developments in radiology is important for ensuring patient safety. The technical advances in the last 20 years that should have lowered radiation exposure (i.e., dose) from intraoral examinations have not occurred due to increased accessibility of X-ray equipment and the use of multiple retakes [3,4]. Studies have shown that frequent bitewing examinations in childhood and adolescence are not necessarily connected to the risk of caries or based on individual indications, but rather as being part of a screening process where the dentists presume the risk of caries is higher than the risks associated with ionizing radiation [5,6].

Ionizing radiation can cause cancer and have negative effects on the eyes and thyroid gland. The radiation dose from dental X-ray examinations is low compared to most medical X-ray examinations. At these low dose levels, there is no clear epidemiological evidence for radiation-induced cancer from single X-ray examinations. However, very low cancer risks are still assumed but indistinguishable from the baseline cancer rate due to insufficient statistical power [7]. This implies the risk of radiation-induced cancer can accumulate with multiple examinations of the same individual or as a collective risk to the population. This understanding is especially important for dental X-rays as they are more frequently used than medical X-rays. The patients exposed to dental X-rays are also generally younger (usually children) and healthier than patients exposed to medical X-rays. Children are more sensitive to ionizing radiation, further increasing the need to understand the risks from dental X-rays.

Some studies conclude that there is a risk for meningioma [8,9] after dental X-rays. These results are based on historical data and on the participants' ability to remember dental X-ray examinations from their childhood. Some studies use results from before 1945 when the doses were much higher than today. Systematic reviews have been done with mixed results, although they are mostly based on the same primary studies. This further shows the gap in knowledge and the need for a different approach where the qualities of the studies are assessed using well-established protocols before making any conclusions.

It is important to have reliable information regarding these risks. The risk with ionizing radiation concern patients. Studies showing potential risks of harmful medical impact may cause disproportionate negative media attention and public trust can be hard to regain.

If the use of X-rays in dentistry entails risks for the development of cancer, it is important that these risks are well described as they may affect patients' willingness to undergo X-ray examinations.

Hence, this study addresses the following research question: What are the effect sizes of negative health effects of dental X-rays among children and adolescents?

## Materials and methods

### Objectives

This study investigates the evidence for health risks associated with dental X-rays. The protocol was registered at PROSPERO International prospective register of systematic reviews (CRD42022369405). PRISMA checklist is presented in in S1 File.

### Eligibility criteria for studies

Eligibility criteria for inclusion of the studies were as follows: a predefined study population with age and sex registered. Population Exposure Control Outcome (PECO) as well as inclusion and exclusion criteria for eligible studies are summarized in Table 1.

### Literature search

A search was performed in the following databases: Medline (Ovid), Embase (embase.com), and Web of Science (Clarivate). The final search was conducted the 15th of August 2024.

### Search strategies

The search strategy was developed in Medline (Ovid) in collaboration with librarians at the Karolinska Institutet University Library. For each search concept, Medical Subject Headings (MeSH terms) and free text terms were identified. The search was then translated into the other databases with assistance from the Polyglot Search Translator [10]. The search was limited to studies written in English and databases were searched from inception. These strategies were reviewed by

**Table 1. PECO and inclusion/exclusion criteria.**

| P | Children and adolescents, exposure at 0–23 of age |
|---|---|
| E | X-ray exposure in dental setting |
| C | Different levels of X-ray exposure, no exposure |
| O | Cancer, eye diseases, effects on thyroid gland |
| **Systematic reviews** | **Inclusion criteria** |
| | Systematic review |
| | Systematic meta-analysis |
| | English abstract |
| | **Exclusion criteria** |
| | Non-systematic review |
| | Guidelines |
| | *Consensus statements* |
| **Primary studies** | **Inclusion criteria** |
| | English abstract |
| | Cohort studies or case control studies |
| | **Exclusion criteria** |
| | Animal studies |
| | *In vitro* studies |

another librarian before execution. De-duplication was done using the method described by Bramer et al. [11]. An extra step was added to compare DOIs. A snowball search was applied to check references and citations of eligible studies from the database searches using EndNoteX9. The full search strategies for all databases are listed in Table 2.

**Table 2. Search strategy.**

| Database | Search strategy |
|---|---|
| Medline<br>Number of hits:5,729 | radiography, dental/ OR radiography, bitewing/OR radiography, dental, digital/ OR radiography, panoramic/ OR ((dental or dentist* or bitewing or bitewing or tooth or teeth or panoramic) adj3 (ct or cbct or cone beam* or radiograph* or radiovisio-graph* or tomograph* or x-ray* or xray*)).ab.ti,kf. OR risk OR risk [MeSH Terms] OR radiation effects OR radiation effects [MeSH Terms] radiation exposure OR radiation exposure [MeSH Terms] OR Dose-Response Relationship, Radiation OR Dose-Response Relationship, Radiation [MeSH Terms] OR risk?.ti,ab,kf. OR (radiation adj3 (effect? or expos* or injur* or induc*)).ti,ab,kf. OR dose response.ti,ab,kf. neoplasm OR neoplasm [MeSH Terms] OR (cancer? or neoplasm? or tumo?r? or cyst?).ti,ab,kf. |
| Embase<br>Number of hits:8,367 | **#19**<br>**#17** NOT **#18**<br>**#18**<br>**#6** AND **#15** AND [english]/lim AND ([conference abstract]/lim OR [conference paper]/lim OR [conference review]/lim)<br>**#17**<br>**#6** AND **#15** AND [english]/lim<br>**#16**<br>**#6** AND **#15**<br>**#15**<br>**#7** OR **#8** OR **#9** OR **#10** OR **#11** OR **#12** OR **#13** OR **#14**<br>**#14** cancer$:ti,ab,kw OR neoplasm$:ti,ab,kw OR tumo$r$:ti,ab,kw OR cyst$:ti,ab,kw<br>**#13**'neoplasm'/exp<br>**#12**'dose response':ti,ab,kw<br>**#11**(**radiation** NEAR/3 (effect$ OR **expos*** OR **injur*** OR **induc***)):ti,ab,kw<br>**#10** risk$:ti,ab,kw<br>**#9** 'radiation exposure'/exp<br>**#8**'radiation response'/exp<br>**#7**'risk'/exp<br>**#6**<br>**#1** OR **#2** OR **#3** OR **#4** OR **#5**<br>**#5**((**dental**:kw OR **dentist***:kw OR **bitewing**:kw OR 'bite wing':kw OR **tooth**:kw OR **teeth**:kw OR **panoramic**:kw) AND (**ct**:kw OR **cbct**:kw OR 'cone beam*':kw OR **radiograph***:kw OR **radiovisiograph***:kw OR **tomograph***:kw OR 'x ray*':kw OR **xray***:kw)<br>**#4** ((**dental**:ti OR **dentist***:ti OR **bitewing**:ti OR 'bite wing':ti OR **tooth**:ti OR **teeth**:ti OR **panoramic**:ti) AND (**ct**:ti OR **cbct**:ti OR 'cone beam*':ti OR **radiograph***:ti OR **radiovisiograph***:ti OR **tomograph***:ti OR 'x ray*':ti OR **xray***:ti)<br>**#3**((**dental** OR **dentist*** OR **bitewing** OR 'bite wing' OR **tooth** OR **teeth** OR **panoramic**) NEAR/3 (**ct** OR **cbct** OR 'cone beam*' OR **radiograph*** OR **radiovisiograph*** OR **tomograph*** OR 'x ray*' OR **xray***)):ab<br>**#2**'panoramic radiography'/de<br>**#1**'tooth radiography'/de |
| Web of Science Core Collection<br>Number of hits:3,953 | 1: AB=((dental OR dentist* OR bitewing OR bite-wing OR tooth OR teeth OR panoramic) NEAR/3 (ct OR cbct OR "cone beam*" OR radiograph* OR radiovisiograph* OR tomograph* OR x-ray* OR xray*))<br>2: TI=((dental OR dentist* OR bitewing OR bite-wing OR tooth OR teeth OR panoramic) AND (ct OR cbct OR "cone beam*" OR radiograph* OR radiovisiograph* OR tomograph* OR x-ray* OR xray*))<br>3: AK=((dental OR dentist* OR bitewing OR bite-wing OR tooth OR teeth OR panoramic) AND (ct OR cbct OR "cone beam*" OR radiograph* OR radiovisiograph* OR tomograph* OR x-ray* OR xray*))<br>4: KP=((dental OR dentist* OR bitewing OR bite-wing OR tooth OR teeth OR panoramic) AND (ct OR cbct OR "cone beam*" OR radiograph* OR radiovisiograph* OR tomograph* OR x-ray* OR xray*))<br>5: #4 OR #3 OR #2 OR #1<br>6: TS=risk$<br>7: TS=(radiation NEAR/3 (effect$ OR expos* OR injur* OR induc*))<br>8: TS="dose response"<br>9: TS=(cancer$ OR neoplasm$ OR tumor$ OR tumour$ OR cyst$)<br>10: #8 OR #7 OR #6 OR #9<br>11: #10 AND #5<br>12: #10 AND #5 and English (Languages) |

## Study selection

The Rayyan software program (Qatar Computing Research Institute; Data Analytics) was used to manage the references and 10,949 records were imported and in a first step duplicates were removed. The retrieved list of publications was subject to a crude exclusion of irrelevant publications based on title. In case of uncertainty, the publication remained included until the next selection step for the assessment of the abstracts. The abstracts were read by six reviewers independently divided into two groups: Aron Naimi-Akbar (ANA), Laila Wiklander (LW) and Nils Kadesjö (NK), in one group. Andreas Cederlund (AC), Sofia Tranaeus (ST), Peggy Näsman (PN), and LW in the other. 10,894 records were excluded after reading the abstracts because they did not meet the inclusion criteria presented in Table 1. Finally, 55 studies were selected for full text reading by all six reviewers (AC, ANA, LW, NK, PN, ST). Any disagreement during the screening process, from abstract to full text, was solved by discussion in the project group. Duplicates, non-relevant studies such as case reports, studies with wrong exposure, book chapters, letters to editor, comments, and studies with nonrelevant outcomes were excluded. All articles excluded (n = 37) are listed in Table 3 with the reason(s) for exclusion. The included articles are divided into systematic reviews (n = 7) and primary studies (n = 11).

## Assessment of risk of bias

**Systematic reviews.** The risk of bias in the included systematic reviews were assessed using ROBIS [12].

**Primary studies.** The risk of bias of the included primary studies was assessed using ROBINS-E (version 2023, June 20) [13]. Before the assessment, the group prepared a list of very important and important confounding factors. In the group of very important confounding factors that can influence the outcome, ionizing medical radiation, radiotherapy, age, and gender were set. Additional important confounding factors were socioeconomic factors, lifestyle habits, genetics, radiation sensitivity, syndrome, disease, profession, environmental toxins, and natural background radiation.

## Data extraction

**Systematic reviews.** No systematic review was eligible for data extraction after assessment of risk of bias by the research group.

**Primary studies.** Data were extracted by AC, ANA, LW, PN and ST from the primary studies regarding population (number of patients/study subjects), study period, (length of follow-up), age, sex, and type of outcome. In October 2023 the extraction was finalized by ANA, ST and LW.

## Certainty of evidence

The certainty of the evidence in the studies was evaluated using the Grading of Recommendations, Assessment, Development, and Evaluation (GRADE) in four steps of evidence grading: high, moderate, low, and very low [14].

## Results

### Literature search and study selection

After the updated literature search in August 2024 a result of 10,949 publications were retrieved. In Table 2 the search strategy is presented for each database. The flow chart of the screening process for the studies is shown in Fig 1.

A total of 55 potential publications were gathered for full text reading after the screening of titles and abstracts by the group. Any publications that were not relevant for the aim of this systematic review were excluded. All the excluded publications are presented in Table 3 with reasons for exclusion.

 

**Table 3. Reasons for exclusion (full text level).**

| Author | Year | Reason for exclusion |
|---|---|---|
| Aps & Scott [16] | 2014 | Wrong outcome |
| Corona et al. [17] | 2012 | Wrong population |
| Gibbs et al. [18] | 1988 | Wrong study design |
| Gibson et al. [19] | 1972 | Wrong population |
| Grufferman et al. [20] | 2009 | Wrong population |
| Hallquist & Nasman [21] | 2001 | Wrong exposure |
| Hedesiu et al. [22] | 2018 | Wrong outcome |
| Jargin [23] | 2016 | Narrative review |
| Jha et al. [24] | 2021 | Wrong study design |
| Pflugbeil et al. [25] | 2011 | Narrative review |
| Scarfe [26] | 2012 | Narrative review |
| Schonfeld et al. [27] | 2011 | Narrative review |
| Umansky et al. [28] | 2008 | Narrative review |
| White [29] | 1984 | Book chapter |
| Author not given [30] | 2019 | Narrative review |
| Author not given [31] | 2012 | Narrative review |
| Kaugars & Page [32] | 1990 | Narrative review |
| Memon et al. [33] | 2007 | Poster |
| Neuberger et al. [34] | 1991 | Wrong population |
| White et al. [35] | 2013 | Letter to editor |
| Lin et al. [36] | 2013 | Wrong population |
| Jorgensen [37] | 2013 | Letter to editor |
| Y. Y. Han et al. [38] | 2012 | Wrong population |
| Wingren et al. [39] | 1997 | Wrong population |
| Dirksen et al. [40] | 2013 | Letter to editor |
| Abt [41] | 2012 | Letter to editor |
| Zhang et al. [42] | 2015 | Wrong population |
| Chaparian & Dehghanzade [43] | 2017 | Wrong study design |
| Wrensch et al. [44] | 2000 | Wrong population |
| Pauwels et al. [45] | 2014 | Wrong study design |
| Longstreth et al. [46] | 2004 | Wrong population |
| Yeh & Chen [47] | 2018 | Wrong study design |
| Preston-Martin & Pogoda [48] | 2003 | Wrong exposure/ study design |
| Auvinen et al. [49] | 2022 | Wrong population |
| Memon et al. [50] | 2010 | Wrong population |
| Hallquist et al. [51] | 1994 | Wrong exposure |
| Chaturvedi et al. [52] | 2020 | Wrong exposure |

## Assessment of risk of bias and data extraction

**Systematic reviews.** All systematic reviews were assessed to have high risk of bias in several of the four domains (Table 4). However, in one of the reviews, Memon et al [53], relevant information was presented, which is set in relation to the present systematic review in the Discussion section.

**Primary studies.** The assessment of risk of bias resulted in eleven primary studies. The evaluation of domain 1(A) concluded that only four of the studies passed. In domain 1(A), the focus is on the confounding factors. As the rest had

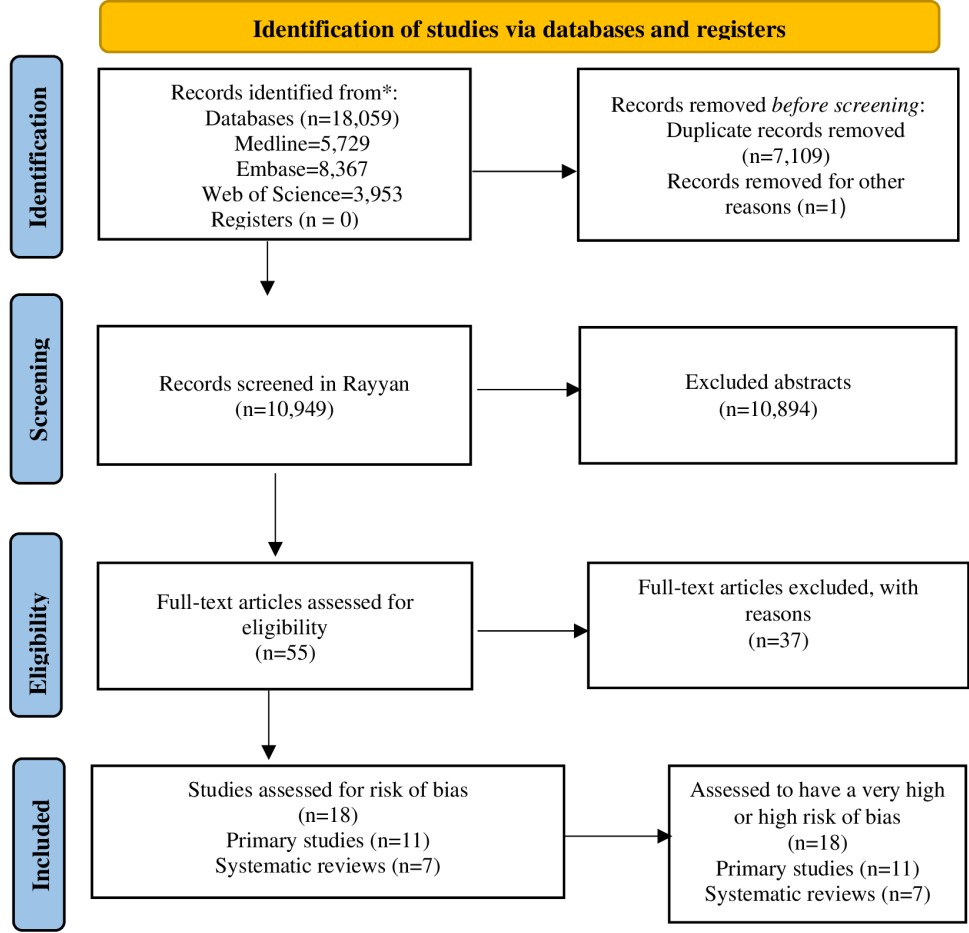

**Fig 1. PRISMA flow chart [ 15].**

a very high risk of bias or high risk of bias, they were excluded (Table 5). In Table 6 all the eleven included studies are presented with type of study population characteristics, type of dental x-ray examination, exposure, outcome, statistical analyses, and comments.

## Meta-analysis

Due to the low quality of evidence for the review question, it was not possible to extract data for a meta-analysis or a Synthesis Without Meta-analysis (SWiM). Additionally, the heterogeneity between the included studies was extremely high.

## Summary of findings

Table 7 summarizes the findings for effects of exposures on meningioma, breast cancer, thyroid cancer, and malignant tumors of the parotid gland. There is very low certainty of evidence for negative health outcomes due to dental x-rays.

## Discussion

This systematic review study investigates evidence for health risks associated with dental X-rays including cancer, eye disease, or effects on the thyroid gland. Eleven primary studies and seven systematic reviews were included. The heterogeneity of the studies made a meta-analysis unsuitable.

**Table 4. Assessment of risk of bias in the systematic reviews.**

| Study | Eligibility Criteria | Identification and Selection | Data Collection and Study Appraisal | Synthesis and Findings | Summary Robis Assess-ments | Additional Assess-ment: Conflicts of Interest | Summary |
|---|---|---|---|---|---|---|---|
| Braganza et al. 2012 [54] | High | High | High | Moderate | High | Low | High risk of bias |
| M.A. Han& Kim 2018 [55] | High | High | High | Moderate | High | Low | High risk of bias |
| Hwang et al. 2018 [56] | High | High | Moderate | Moderate | High | High | High risk of bias |
| Marcu et al. 2021 [57] | High | High | Moderate | Moderate | High | High | High risk of bias |
| Memon et al. 2019 [53] | Moderate | High | Moderate | Moderate | Moderate | Low | Moderate/ High risk of bias |
| Mupparapu et al. 2019 [58] | High | High | High | High | High | High | High risk of bias |
| Xu et al. 2015 [59] | High | High | Moderate | Moderate | Moderate | High | High risk of bias |

**Table 5. Assessment of risk of bias in the primary studies.**

| Study | Risk of bias due to con-founding factors Domain 1A | Risk of bias arising from measurement of the exposure Domain 2C | Summary Domain 1A and 2C |
|---|---|---|---|
| Claus et al. 2012 [8] | Some concerns | Very high | Very high risk of bias |
| Ma et al. 2008 [60] | Some concerns | Very high | Very high risk of bias |
| Khan et al.2010 [61] | Very high | Very high | Very high risk of bias |
| Net et al. 2013 [62] | Low | High | High risk of bias |
| Ryan et al. 1992 [63] | Very high | Very high | Very high risk of bias |
| Preston-Martin et al. 1988 [64] | Low | High | High risk of bias |
| Preston-Martin et al. 1989 [9] | Very high | Very high | Very high risk of bias |
| Preston-Martin et al. 1985 [65] | Very high | Very high | Very high risk of bias |
| Preston-Martin et al. 1983 [66] | Very high | Very high | Very high risk of bias |
| Preston-Martin et al. 1983 [67] | Very high | Very high | Very high risk of bias |
| Preston-Martin et al. 1980 [68] | Very high | Very high | Very high risk of bias |

The included systematic reviews were assed to have a very high or high risk of bias in most cases due to incomplete literature search, such as less than two data bases were searched and/or less than two data extractors. Also, the disclosure of conflict of interest were not presented in many studies. Memon et al. 2019 [53], systematic review presented some evidence for a risk of thyroid cancer due to dental radiography examinations but after the assessment by ROBIS for the identification and selection domain, with the result of a high risk of bias as presented in Table 4.

The literature search is a critical and crucial part of a systematic review that can result in a too narrow search field and/or limitation of keywords if executed careless. For the search and in the retrieving of publications, information specialists were therefore consulted. Our research group included experts in different fields, not only in dentistry for this review, but also an expert within Radiation physics and experts in Health Technology Assessment. But still there can be weaknesses in our study due to misjudgments during the screening process and assessments.

**Table 6. Characteristics of included primary studies.**

| Study | Study design | Study population | Type of dental X-ray | Outcome | Exposure | Results Effect measure (95% CI) | | Statistical model Confounding variables Comments |
|---|---|---|---|---|---|---|---|---|
| Claus et al. 2012 United States | Case-control | 1,433 20–79 years F1049/M384 No of case Control 1350 | BW, FMS, PAN | Meningioma | **Frequency of BW** | **Odds ratio (95% CI)** | | Conditional logistic regression Age, sex, race, education, income, history of head CT Divided into periods; <10 years, 10–19 years, 20–49 years, >50 years. Self-reporting |
| | | | | | <10 years | | | |
| | | | | | None | 1.0 | | |
| | | | | | Less than yearly | 1.3 (1.0–1.8) | | |
| | | | | | Yearly or more | 1.4 (1.0–1.8) | | |
| | | | | | 10–19 years | | | |
| | | | | | None | 1.0 | | |
| | | | | | Less than yearly | 1.3(1.1–1.6) | | |
| | | | | | Yearly or more | 1.6(1.2–2.0) | | |
| | | | | | **Frequency of FMS** | | | |
| | | | | | <10 years | | | |
| | | | | | None | 1.0 | | |
| | | | | | Less than yearly | 1.1(0.9–1.4) | | |
| | | | | | Yearly or more | 1.2(0.9–1.8) | | |
| | | | | | 10–19 years | | | |
| | | | | | None | 1.0 | | |
| | | | | | Less than yearly | 1.1(0.9–1.4) | | |
| | | | | | Yearly or more | 1.2(0.9–1.8) | | |
| | | | | | **Frequency of PAN** | | | |
| | | | | | <10 years | | | |
| | | | | | Ever | 4.9 (1.8–13.2) | | |
| | | | | | 10–19 years | | | |
| | | | | | None | 1.0 | | |
| | | | | | Less than yearly | 1.3(0.9–1.9) | | |
| | | | | | Yearly or more | 3.0(1.2–7.8) | | |
| Ma et al. 2008 United States | Case-control | 1,742 20–49 years No of case Control 441 Only females | Any dental X-ray not specified. Using lead-apron (+) or not (-) | Breast cancer | **Frequency of dental-ray by lead apron (+/-) use** | **Odds ratio (95% CI)** | | Multivariate unconditional logistic regression Age, race, medical X-ray, (radiation therapy), education, BMI-index, alcohol intake, health, and medical history such as parity, hormone therapy use, cancer history in family, demographics, etc. Self-reporting |
| | | | | | <20 years | | | |
| | | | | | Less than once every 5 years (+) | 1.0 | | |
| | | | | | At least once every 5 years (+) | 1.23 (0.76–2.0) | | |
| | | | | | At least once every 3 years (+) | 1.4 (0.97–2.02) | | |
| | | | | | About once a year | 1.03 (0.72–1.47) | | |
| | | | | | Who did not always wear lead apron | 1.81 (1.13–2.90) | | |
| | | | | | Lead apron use unknown | 1.24 (0.83–1.86) | | |
| | | | | | Did not always wear before age of 20 years | 1.52 (0.94–2.47) | | |

*(Continued)*

**Table 6.** (Continued)

| Study | Study design | Study population | Type of dental X-ray | Outcome | Exposure | Results Effect measure (95% CI) | Statistical model Confounding variables Comments |
|---|---|---|---|---|---|---|---|
| Khan et al. 2010 United States | Case-control | 318 <6 years No information about gender No of case controls 318 | Standard dental X-ray not specified number of intraoral X-rays. PAN | Medullo-blastoma and primitive neuroecto-dermal tumour | 7 cases had PAN examinations. Single dental x-rays count as exposure | **Odds ratio (95% CI)** 0.85 (0.37–1.9) | Conditional logistic regression Mother´s race, age, sex, caries, income, mother´s education, mothers' marital status, mother´s income, behavioural &, contex-tual variables. Maternal recall/reporting |
| Neta et al. 2013 United States | Cohort | 251 22–86 years F219, M32 75,234 | BW, FMS, PAN | Thyroid cancer | Increased risk for every 10 dental X-rays | **Hazard ratios (95% CI)** 1.13 (1.01–1.26) | Multivariate Cox proportional hazard regression Age, race, smoking status, BMI-index, medical history estimated occupational radiation dose to the thyroid- Radiologic Technologists Self-reporting |
| Ryan et al. 1992 Australia | Case-control | 170 25–74 years F/M no of each gender not specified. Control 417 | FMS, PAN | Glioma | Exposure to at least 1 PAN and FMS<25 years | **Risk ratio (95% CI)** 0.42 (0.19–0.93) | Unconditional multiple logistic regression Adjusted for age, sex, socio-economic status, smoking, alcohol, lifestyle habits, medical history, exposure to other forms of radiation. Self-reporting |
| | | | | Meningioma | Exposure to at least 1 PAN and FMS<25 years | **Risk ratio (95% CI)** 0.49 (0.16–1.54) | |
| Preston-Martin et al. 1988 | Case-control | 139 25–64 years F74, M65 Con-trol 139 | FMS, PAN, Cepha-lometric examinations | Tumours of parotid gland malignant | **Exposure dose (Rad)** | **Cumulative parotid dose from dental radiography, Risk ratio (95%CI)** | No results/reports by age for exposure, Unadjusted for age, medical X-ray, Exposure before 1979 |
| | | | | | 5–24.9 | 1.2 (0.65–2.29) | |
| | | | | | 25–49.9 | 1.7 (0.5–5.68) | |
| | | | | | >50 | 2.8 (0.85–9.48) | |
| Preston-Martin et al. 1989 | Case-control | 202 Glioma 70 Meningioma (Only men) 25–69 Controls 272 | Any dental X-ray. FMS | Glioma | Frequency of dental X-rays up to age 25 | **Odds ratio (95% CI)** | Unadjusted |
| | | | | | Every 2–5 years | 1.5 (1.0–2.4) | |
| | | | | | Once a year | 1.4 (0.8–2.4) | |
| | | | | Meningioma | Frequency of dental X-rays up to age 25 | **Odds ratio (95% CI)** | |
| | | | | | Every 2–5 years | 1.1 (0.5–2.2) | |
| | | | | | Once a year | 1.5 (0.5–4.4) | |
| Preston-Martin et al. 1985 | Case-control | 101 Women 18–64 years. Matched pairs for control | FMS | Meningioma | Had first full mouth dental X-rays under the age of 20 | **Odds ratio** 4.0 | Some unadjusted. some adjusted Based on other studies reus-ing epidemiological data from already presented studies. Case-control study of intracranial meningiomas in Women in Los Angeles County. California 1980. Risk factors for meningiomas in Men in Los Angeles County 1983(a); Epidemiology of intra-cranial Meningiomas: Los Ange-les County. California 1983(b) |
| | | 105 25–69 years Only males Controls 105 | FMS | Meningioma | Had five or more full mouth dental X-rays before 1945. | 2.7 | |
| | | 304 paired controls | FMS | Tumours of parotid gland malignant | More 5 or more before 1960 (age at exposure unknown) | 4.3 | |

*(Continued)*

**Table 6.** (Continued)

| Study | Study design | Study population | Type of dental X-ray | Outcome | Exposure | Results Effect measure (95% CI) | Statistical model Confounding variables Comments |
|---|---|---|---|---|---|---|---|
| Preston-Martin et al. 1983 (a) | Case-control | 105 25–69 years Only males Controls 105 | FMS | Meningioma | Had five or more full mouth dental X-rays before 1945. | **Odds ratio** 2.7 | Unadjusted |
| Preston-Martin et al. 1983 (b) | Case-control | W 185 with paired controls W 18–64 years M 105 with paired controls M 25–69 years | FMS | Meningioma | Had first full mouth dental X-rays under the age of 20 | **Odds ratio** 4.0 (Women) 1.0 (Men) | Adjusted for matches with sex, age, race, socioeconomics. Interviewed women's relatives if deceased. Men only those alive-recall bias. Exposure before 1945 |
| Preston-Martin et al. 1980 | Case-control | 101 Women 18–64 years Matched pairs for control | FMS | Meningioma | Had first full mouth dental X-rays under the age of 20 | **Odds ratio** 4.0 | Some adjustments for confounding factors. Recall bias |

BW=bitewing, FMS=Full Mouth Series, PAN=Panoramic, Panorex

**Table 7.** Summary of findings for effects of exposure on meningioma, breast cancer, thyroid cancer, and malignant tumours of the parotid gland.

| Exposure Reference | Outcome | Number of subjects (studies) | Results | Certainty of the evidence (GRADE) | Reason for grading down |
|---|---|---|---|---|---|
| Bitewing. full-mouth series, panoramic X-ray, cephalogram Claus et al. 2012 Ma et al. 2008 Neta et al. 2013 Preston-Martin et al. 1988 | Meningioma, breast cancer, thyroid cancer, malignant tumours of the parotid gland | 3565 (4) | – | Very low +OOO | Risk of bias -2[a] Inconsistency -1[b] |
| DT in dental settings, CBCT, or other extra oral projections | Leukaemia, eye diseases, brain cancer | – | – | – | No relevant studies identified |

[a]Weaknesses in study design and statistical analysis.

[b]Inconsistency in the timings, outcome measures, and results between the studies

Some of the included studies are old and based their results on material from the first half of the 20[th] century with the normal settings for the dental radiographic examinations at that time. Preston-Martin et al [65–68] reused some material for more than one study, which has been commented in Table 6. In the same review, more than five studies were included, which decrease the overall certainty due to high risk of reporting bias. The lack of reporting confounding factors that are considered as very important confounders such as ionizing medical radiation, radiotherapy, age and gender for the presented outcome (Table 1); such as cancer, eye diseases, effects on thyroid gland with recall bias makes these studies less reliable. The age of the studies can matter in some of the weaknesses. Thus, to present results that did not pass the quality assessments in a meta-analysis or Synthesis Without Meta-analysis can give the impression of being accurate and reliable and must be avoided for further misinterpretations.

This systematic review shows that the studies included exhibit several weaknesses. Therefore, more studies are necessary to create knowledge based on verified data and contemporary digital techniques. The dental care system is responsible for keeping exposure to x-ray radiation as low as possible (i.e., avoiding overexposure) as dictated by the ALARA (As

Low As Reasonably Archivable) principle. Because dental X-ray examinations are frequent, often from a young age, more accurate information from digital records is needed to keep recall bias to a minimum. Such accurate information can be used to discover whether dental X-rays have negative effects on health.

The estimated individual radiation dose may be widely inaccurate if medical radiation exposures are not considered. Some medical exposures, such as from brain CT, could result in radiation doses that are one or several orders of magnitude higher than from dental examinations. Furthermore, it is possible that some conditions require repeated use of both dental and medical X-ray examinations. Thus, failure to account for medical exposures could result in a systematic overestimation of the risks from dental X-ray examinations.

Another known bias is reverse causation between a cancer and radiological examination. It is possible that an x-ray examination was performed because of symptoms from a developing cancer, even though the cancer was not diagnosed until a later examination. An increased cancer incidence has been shown a short time after X-ray examinations, even though the timespan was too short for the cancer to develop [69]. Therefore, an exclusion period, based on the cancer development rate, should be employed before a cancer diagnosis is made.

Future studies must include much larger cohorts and be register-based to avoid the risk of recall bias. Confounding factors that can influence the outcome must also be weighed in. Collaborations with other research groups can be a possible way for more statistical power and accurate data and keeping the risk of reporting bias low.

The technical advances in health and dental care systems with digitalized records makes it possible to follow children from birth up to 18 years of age and assess the exposures of ionizing radiation. Because of the higher accessibility to radiographic examinations, especially in many developed countries, it is important to include all examinations of ionizing radiation even those with low dose such as intraoral radiographs.

Future studies are therefore needed to assess the risk of long-term effects of dental radiography.

In the medical health care, there are joined systems for radiographic examinations that can be accessed by private and public clinics and hospitals. It would be of high value if dental clinics could have the same possibilities to avoid overexposures especially of children and adolescents that are more sensitive to ionizing radiation.

## Conclusion

We were unable to establish evidence for negative health effects of dental X-rays among children and adolescents. Our findings highlight the fact that there is a need for new studies with thorough study protocols, including number of subjects, verified data, and adjustments for predefined confounding factors to improve scientific knowledge.

## Supporting information

**S1 File. PRISMA_2020_checklist.**
(DOCX)

**S1 Table. With 10949 included records.**
(XLSX)

**S2 Table. With 11 included primary studies.**
(XLSX)

## Acknowledgments

The authors would like to acknowledge Emma-Lotta Säätelä & Sabina Gillsund, Karolinska Institutet University Library, for skillful assistance, and Hossein Ordoubadian, Accent Språkservice for language revision.

## Author contributions

**Conceptualization:** Laila Wiklander, Andreas Cederlund, Nils Kadesjö, Sofia Tranaeus, Aron Naimi-Akbar.

**Data curation:** Laila Wiklander, Nils Kadesjö, Sofia Tranaeus, Aron Naimi-Akbar.

**Formal analysis:** Nils Kadesjö, Sofia Tranaeus, Aron Naimi-Akbar.

**Funding acquisition:** Laila Wiklander.

**Investigation:** Laila Wiklander, Andreas Cederlund, Nils Kadesjö, Peggy Näsman, Sofia Tranaeus, Aron Naimi-Akbar.

**Methodology:** Laila Wiklander, Nils Kadesjö, Peggy Näsman, Sofia Tranaeus, Aron Naimi-Akbar.

**Project administration:** Laila Wiklander, Sofia Tranaeus, Aron Naimi-Akbar.

**Resources:** Laila Wiklander, Andreas Cederlund, Sofia Tranaeus, Aron Naimi-Akbar.

**Software:** Sofia Tranaeus, Aron Naimi-Akbar.

**Supervision:** Andreas Cederlund, Nils Kadesjö, Peggy Näsman, Sofia Tranaeus, Aron Naimi-Akbar.

**Validation:** Laila Wiklander, Andreas Cederlund, Nils Kadesjö, Peggy Näsman, Sofia Tranaeus, Aron Naimi-Akbar.

**Visualization:** Laila Wiklander, Andreas Cederlund, Nils Kadesjö, Sofia Tranaeus, Aron Naimi-Akbar.

**Writing – original draft:** Laila Wiklander, Andreas Cederlund, Nils Kadesjö, Sofia Tranaeus, Aron Naimi-Akbar.

**Writing – review & editing:** Laila Wiklander, Andreas Cederlund, Nils Kadesjö, Peggy Näsman, Sofia Tranaeus, Aron Naimi-Akbar.

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
