## [Decision Letter · Decision Letter 0]

12 Feb 2025

PONE-D-24-37898Negative health effects of dental x-rays: a systematic reviewPLOS ONE

Dear Dr. Wiklander,

Thank you for submitting your manuscript to PLOS ONE. After careful consideration, we feel that it has merit but does not fully meet PLOS ONE’s publication criteria as it currently stands. Therefore, we invite you to submit a revised version of the manuscript that addresses the points raised during the review process.

Dear Authors,

Kindly read all the comments given by the reviewers carefully and address them; make the changes in the revised manuscript accordingly.

Best regards and keep well

We look forward to receiving your revised manuscript.

Kind regards,

Mohmed Isaqali Karobari, BDS, MScD.Endo, Ph.D. Endo, FDS, FPFA, MFDS

Academic Editor

PLOS ONE

- https://doi.org/10.1080/00016357.2020.180704

In your revision ensure you cite all your sources (including your own works), and quote or rephrase any duplicated text outside the methods section. Further consideration is dependent on these concerns being addressed.

3. Thank you for stating the following financial disclosure:  [The study is funded by Folktandvården/Public Dental Health Care, Stockholms Län AB, Region Stockholm].

4. As required by our policy on Data Availability, please ensure your manuscript or supplementary information includes the following:

Additional Editor Comments:

Dear Authors,

Kindly read all the comments given by the reviewers carefully and address them; make the changes in the revised manuscript accordingly.

Best regards and keep well

Reviewers' comments:

Reviewer's Responses to Questions

**Comments to the Author**

1. Is the manuscript technically sound, and do the data support the conclusions?

Reviewer #1: Yes

Reviewer #2: Yes

2. Has the statistical analysis been performed appropriately and rigorously? 

Reviewer #1: Yes

Reviewer #2: Yes

3. Have the authors made all data underlying the findings in their manuscript fully available?

Reviewer #1: Yes

Reviewer #2: Yes

4. Is the manuscript presented in an intelligible fashion and written in standard English?

Reviewer #1: Yes

Reviewer #2: Yes

5. Review Comments to the Author

Reviewer #1: The study is well designed and appropriate consideration has to be given to not perform meta-analysis given the heterogenity of the included studies. The authors could have zeroed in on key factors that relate to negative factors as non eof the key words have emphasized - "adverse effects". Nevertheless, the study is well framed

Reviewer #2: 1.Keywords: Keywords as per Mesh terms needed. Kindly search keywords on MeSH on Demand, The MeSH Browser & Direct Browsing of MeSH Hierarchy (Trees). Use appropriate keywords applying Boolean terms (Please use keywords as per MESH headings on Medline)

2. Incomplete Discussion section & needs to extensively elaborated

3. Limitations of this review needs to be mentioned.

4. Future scope of the review based on the outcomes of total number of studies synthesized needs to be mentioned

6. PLOS authors have the option to publish the peer review history of their article (what does this mean? ). If published, this will include your full peer review and any attached files.

**Do you want your identity to be public for this peer review?** For information about this choice, including consent withdrawal, please see our Privacy Policy .

Reviewer #1: **Yes: ** Sudhir Rama Varma

Reviewer #2: No

---

## [Author Response · Author response to Decision Letter 1]

8 Apr 2025

Thank you so much for your comments to our work! Hopefully we have met your requests succesfully. I have uploaded the file. Kindest regards Laila

---

## [Decision Letter · Decision Letter 1]

15 Apr 2025

Negative health effects of dental x-rays: a systematic review

PONE-D-24-37898R1

Dear Dr. Wiklander,

We’re pleased to inform you that your manuscript has been judged scientifically suitable for publication and will be formally accepted for publication once it meets all outstanding technical requirements.

Kind regards,

Mohmed Isaqali Karobari, BDS, MScD.Endo, Ph.D. Endo, FDS, FPFA, MFDS

Academic Editor

PLOS ONE

Additional Editor Comments (optional):

Dear Authors,

The authors have addressed all the comments and suggestions reviewers gave, and the manuscript has dramatically improved. The manuscript can be accepted for publication in its current form. I would like to congratulate the authors and wish them all the very best in their future endeavors.

Best regards and keep well.

Reviewers' comments:

Reviewer's Responses to Questions

**Comments to the Author**

1. If the authors have adequately addressed your comments raised in a previous round of review and you feel that this manuscript is now acceptable for publication, you may indicate that here to bypass the “Comments to the Author” section, enter your conflict of interest statement in the “Confidential to Editor” section, and submit your "Accept" recommendation.

Reviewer #2: All comments have been addressed

2. Is the manuscript technically sound, and do the data support the conclusions?

Reviewer #2: Yes

3. Has the statistical analysis been performed appropriately and rigorously? 

Reviewer #2: Yes

4. Have the authors made all data underlying the findings in their manuscript fully available?

Reviewer #2: Yes

5. Is the manuscript presented in an intelligible fashion and written in standard English?

Reviewer #2: Yes

6. Review Comments to the Author

Reviewer #2: I appreciate the authors efforts for addressing all the comments well. I appreciate the efforts put in conducting the elaborate research and manuscript writing. The manuscript is well written and explains a relevant topic of research.

7. PLOS authors have the option to publish the peer review history of their article (what does this mean? ). If published, this will include your full peer review and any attached files.

**Do you want your identity to be public for this peer review?** For information about this choice, including consent withdrawal, please see our Privacy Policy .

Reviewer #2: No

---

## [Editor Report · Acceptance letter]

PONE-D-24-37898R1

PLOS ONE

Dear Dr. Wiklander,

I'm pleased to inform you that your manuscript has been deemed suitable for publication in PLOS ONE. Congratulations! Your manuscript is now being handed over to our production team.

Kind regards,

on behalf of

Prof Dr. Mohmed Isaqali Karobari

Academic Editor

PLOS ONE